# Implicit Deep Adaptive Design: Policy–Based Experimental Design without Likelihoods

**Desi R. Ivanova**[†] **Adam Foster**[†] **Steven Kleinegesse**[‡] **Michael U. Gutmann**[‡] **Tom Rainforth**[†]

[†]Department of Statistics, University of Oxford
[‡] School of Informatics, University of Edinburgh

desi.ivanova@stats.ox.ac.uk

## Abstract

We introduce implicit Deep Adaptive Design (iDAD), a new method for performing adaptive experiments in *real-time* with *implicit* models. iDAD amortizes the cost of Bayesian optimal experimental design (BOED) by learning a design *policy network* upfront, which can then be deployed quickly at the time of the experiment. The iDAD network can be trained on any model which simulates differentiable samples, unlike previous design policy work that requires a closed form likelihood and conditionally independent experiments. At deployment, iDAD allows design decisions to be made in milliseconds, in contrast to traditional BOED approaches that require heavy computation during the experiment itself. We illustrate the applicability of iDAD on a number of experiments, and show that it provides a fast and effective mechanism for performing adaptive design with implicit models.

## 1 Introduction

Designing experiments to maximize the information gathered about an underlying process is a key challenge in science and engineering. Most such experiments are naturally *adaptive*—we can design later iterations on the basis of data already collected, refining our understanding of the process with each step [36, 45, 51]. For example, suppose that a chemical contaminant has accidentally been released and is rapidly spreading; we need to quickly discover its unknown source. To this end, we measure the contaminant concentration level at locations $\xi_1, \ldots, \xi_T$ (our experimental designs), obtaining observations $y_1, \ldots, y_T$. Provided we can perform the necessary computations sufficiently quickly, we can design each $\xi_t$ using data from steps $1, \ldots, t-1$ to narrow in on the source.

Bayesian optimal experimental design (BOED) [7, 32] is a principled model-based framework for choosing designs optimally; it has been successfully adopted in a diverse range of scientific fields [52, 58, 60]. In BOED, the unknown quantity of interest (e.g. contaminant location) is encapsulated by a parameter $\theta$, and our initial information about it by a prior $p(\theta)$. A simulator, or likelihood, model $y|\theta, \xi$ describes the relationship between $\theta$, our controllable design $\xi$, and the experimental outcome $y$. To select designs *optimally*, the guiding principle is *information maximization*—we select the design that maximizes the expected (Shannon) information gained about $\theta$ from the data $y$, or, equivalently, that maximizes the mutual information between $\theta$ and $y$.

This naturally extends to adaptive settings by considering the *conditional* expected information gain given previously collected data. The traditional approach, depicted in Figure 1a, is to fit a posterior $p(\theta|\xi_{1:t-1}, y_{1:t-1})$ after each iteration, and then select $\xi_t$ in a myopic fashion using the one-step mutual information (see, e.g., [51] for a review). Unfortunately, this approach necessitates significant computation at each $t$ and does not lend itself to selecting optimal designs quickly and adaptively.

Recently, Foster et al. [17] proposed an exciting alternative approach, called Deep Adaptive Design (DAD), that is based on learning design *policies*. DAD provides a way to avoid significant computation

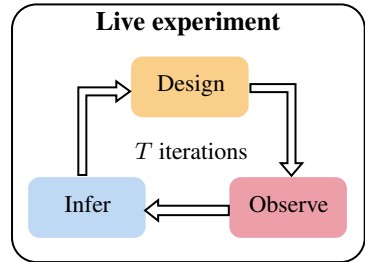

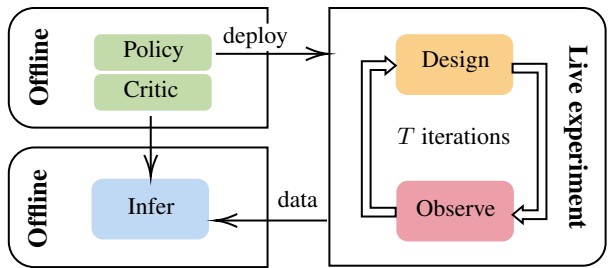

(a) Traditional BOED: costly computations (design optimisation and parameter inference) are required at each iteration.

(b) Policy-based BOED using iDAD: a design policy and critic are learnt before the live experiment. The policy enables quick and adaptive experiments, the critic assists likelihood-free inference.

Figure 1: Overview of adaptive BOED approaches applicable to implicit models.

at deployment-time by, prior to the experiment itself, learning a design policy network that takes past design-outcome pairs and near–instantaneously returns the design for the next stage of the experiment. The required training is done using simulated experimental histories, without the need to estimate any posterior or marginal distributions. DAD further only needs a single policy network to be trained for multiple experiments, further allowing for *amortization* of the adaptive design process. Unfortunately, DAD requires conditionally independent experiments and only works for the restricted class of models that have an explicit likelihood model we can simulate from, evaluate the density of, and calculate derivatives for, substantially reducing its applicability.

To address this shortfall, we instead consider a far more general class of models where we require only the ability to simulate $y|\theta, \xi$ and compute the derivative $\partial y/\partial \xi$, e.g. via automatic differentiation [5]. Such models are ubiquitous in scientific modelling and include differentiable *implicit models* [19], for which the likelihood density $p(y|\theta, \xi)$ is intractable. Examples include mixed effects models [15, 18], various models from chemistry and epidemiology [1], the Lotka Volterra model used in ecology [19], and models specified via stochastic differential equations (such as the SIR model [10]).

To perform rapid adaptive experimentation with this large class of models, we introduce *implicit Deep Adaptive Design* (iDAD), a method for learning adaptive design policy networks using only simulated outcomes (see Figure 1b). To achieve this, we introduce likelihood-free lower bounds on the total information gained from a sequence of experiments, which iDAD utilizes to learn a deep policy network. This policy network amortizes the cost of experimental design for implicit models and can be run in milliseconds at deployment-time. To train it, we show how the InfoNCE [57] and NWJ [37] bounds, popularized in representation learning, can be applied to the policy-based experimental design setting. The optimization of both of these bounds involves simultaneously learning an auxiliary *critic network*, bringing an important added benefit: it can be used to perform likelihood-free posterior inference of the parameters given the data acquired from the experiment.

We also relax DAD's requirement for experiments to be conditionally independent, allowing its application in complex settings like time series data, and, through innovative architecture adaptations, also provide improvements in the conditionally independent setting as well. This further expands the model space for policy-based BOED, and leads to additional performance improvements.

Critically, iDAD forms the first method in the literature that can practically perform real-time adaptive BOED with implicit models: previous approaches are either not fast enough to run in real-time for non-trivial models, or require explicit likelihood models. We illustrate the applicability of iDAD on a range of experimental design problems, highlighting its benefits over existing baselines, even finding that it often outperforms costly non-amortized approaches. Code for iDAD is publicly available at https://github.com/desi-ivanova/idad.

## 2   Background

The BOED framework [32] begins by specifying a Bayesian model of the experimental process, consisting of a prior on the unknown parameters $p(\theta)$, a set of controllable designs $\xi$, and a data generating process that depends on them $y|\theta, \xi$; as usual in BOED, we assume that $p(\theta)$ does not depend on $\xi$. In this paper, we consider the situation where $y|\theta, \xi$ is specified *implicitly*. This means

that it is defined by a deterministic transformation, $f(\varepsilon; \theta, \xi)$, of a base (or noise) random variable, $\varepsilon$, that is independent of the parameters and the design; e.g., $\varepsilon \sim \mathcal{N}(\varepsilon; 0, I)$. The function $f$ is itself often not known explicitly in closed form, but is implemented as a stochastic computer program (i.e. simulator) with input $(\theta, \xi)$ and $\varepsilon$ corresponding to the draws from the underlying random number generator (or equivalently the random seed). Regardless, the resulting induced likelihood density $p(y|\theta, \xi)$ is still generally intractable, but sampling $y|\theta, \xi$ is possible.

Having acquired a design-outcome pair $(\xi, y)$, we can quantify the amount of information we have gained about $\theta$ by calculating the reduction in entropy from the prior to the posterior. We can further assess the quality of a design $\xi$ before acquiring $y$, by computing the expected reduction in entropy with respect to the marginal distribution of the outcome, $p(y|\xi) = \mathbb{E}_{p(\theta)}[p(y|\theta, \xi)]$. The resulting quantity, called the *expected information gain* (EIG), is of central interest in BOED and is defined as

$$I(\xi) := \mathbb{E}_{p(\theta)p(y|\theta,\xi)} \left[ \log \frac{p(\theta|\xi, y)}{p(\theta)} \right] = \mathbb{E}_{p(\theta)p(y|\theta,\xi)} \left[ \log \frac{p(y|\theta, \xi)}{p(y|\xi)} \right]. \tag{1}$$

Note that $I(\xi)$ is equivalent to the mutual information (MI) between the parameters $\theta$ and data $y$ when performing experiment $\xi$. The optimal $\xi$ is then the one that maximises the EIG, i.e. $\xi^* = \arg\max_\xi I(\xi)$. Performing this optimization is a major computational challenge since the information objective is doubly intractable [46]. For implicit models, the cost becomes even greater as the likelihood is also not available in closed form, so estimating it, along with the marginal likelihood $p(y|\xi)$, is already itself a major computational problem [11, 20, 33, 54].

Jointly optimizing the design variables for all undertaken experiments at the same time using (1) is called *static* experimental design. In practice, however, we are often more interested in performing multiple experiments *adaptively* in a sequence $\xi_1, \ldots, \xi_T$, so that the choice of each $\xi_t$ can be guided by past experiments, namely the corresponding *history* $h_{t-1} := \{(\xi_i, y_i)\}_{i=1:t-1}$. The typical approach in such settings is to sequentially perform (approximate) posterior inference for $\theta|h_{t-1}$, followed by a one-step look ahead (myopic) BOED optimization that conditions on the observed history. In other words, to determine the designs $\xi_1, \ldots, \xi_T$, we sequentially optimize the objectives

$$I_{h_{t-1}}(\xi_t) := \mathbb{E}_{p(\theta|h_{t-1})p(y_t|\theta,\xi,h_{t-1})} \left[ \log \frac{p(y_t|\theta, \xi, h_{t-1})}{p(y_t|\xi, h_{t-1})} \right], \quad t = 1, \ldots, T. \tag{2}$$

However, such approaches incur significant computational cost during the experiment itself, particularly for implicit models [16, 21, 30]. This has critical consequences: in most cases they cannot be run in real-time, undermining one's ability to use them in practice.

## 2.1 Policy-based adaptive design with likelihoods

For tractable likelihood models, Foster et al. [17] proposed a new framework, called Deep Adaptive Design (DAD), for adaptive experimental design that avoids expensive computations during the experiment. To achieve this, they introduce a parameterized deterministic design function, or policy, $\pi_\phi$ that takes the history $h_{t-1}$ as input and returns the design $\xi_t = \pi_\phi(h_{t-1})$ to be used for the next experiment as output. This set-up allows them to consider the objective

$$\mathcal{I}_T(\pi_\phi) = \mathbb{E}_{p(\theta)p(h_T|\theta,\pi_\phi)} \left[ \sum_{t=1}^{T} I_{h_{t-1}}(\xi_t) \right], \quad \xi_t = \pi_\phi(h_{t-1}), \tag{3}$$

which crucially depends on the policy $\pi$ rather than the individual design $\xi_t$. Learning a policy up-front, rather than designs, is what allows adaptive experiments to be performed in real-time.

Under the assumption that $y_t$ is independent of $h_{t-1}$ conditional on the parameters $\theta$ and the design $\xi_t$, i.e. $p(y_t|\theta, \xi_t, h_{t-1}) = p(y_t|\theta, \xi_t)$, Foster et al. [17] showed that the objective can be simplified to

$$\mathcal{I}_T(\pi_\phi) = \mathbb{E}_{p(\theta)p(h_T|\theta,\pi_\phi)} \left[ \log \frac{p(h_T|\theta, \pi_\phi)}{p(h_T|\pi_\phi)} \right], \quad p(h_T|\theta, \pi_\phi) = \prod_{t=1}^{T} p(y_t|\theta, \xi_t). \tag{4}$$

To deal with the marginal $p(h_T|\pi_\phi)$ in the denominator, they then derived several optimizable lower bounds on $\mathcal{I}_T(\pi_\phi)$, such as the sequential Prior Contrastive Estimation (sPCE) bound

$$\mathcal{L}_T^{\text{sPCE}}(\pi_\phi, L) = \mathbb{E}_{p(\theta_0)p(h_T|\theta,\pi_\phi)p(\theta_{1:L})} \left[ \log \frac{p(h_T|\theta_0, \pi_\phi)}{\frac{1}{L+1} \sum_{\ell=0}^{L} p(h_T|\theta_\ell, \pi_\phi)} \right] \leq \mathcal{I}_T(\pi_\phi) \ \forall L \geq 1. \tag{5}$$

The parameters of the policy $\phi$, which takes the form of a deep neural network, are now learned prior to the experiment(s) using stochastic gradient ascent on this bound with simulated experimental histories. Design decisions can then be made using a single forward pass of $\pi_\phi$ during deployment. Unfortunately, training the DAD network by optimizing (5) requires the likelihood density $p(h_T|\theta,\pi)$ to be analytically available—an assumption that is too restrictive in many practical situations. The architecture for DAD also assumes conditionally independent designs, which is unsuitable in some settings like time-series data. Our method lifts both of these restrictions.

## 3  Implicit Deep Adaptive Design

We have seen that the traditional step-by-step approach to adaptive design for implicit models [16, 21, 30] is too costly to deploy for most applications, whilst the only existing policy-based approach, DAD [17], makes restrictive assumptions that prevent it being applied to implicit models. We aim to relax the restrictive assumptions of the latter, making policy-based BOED applicable to all models where we can sample from $y|\theta,\xi$ and compute the derivative $\partial y/\partial\xi$, a strict superset of the class of models that can be handled by DAD. This requires new training objectives for the policy network that do not involve an explicit likelihood and are not based on conditionally independent experiments, along with new architectures that work for non-exchangeable models like time series.

### 3.1  Information lower bounds for policy-based experimental design without likelihoods

To establish a suitable likelihood-free training objective for the implicit setting, our high-level idea is to leverage recent advances in variational MI [see 42, for an overview], which have shown promise for *static* BOED [16, 28, 29]. While using these bounds in the traditional framework of (2) would not permit real-time experiments, one could consider a naive application of them to the policy objective of (3) by replacing each $I_{h_{t-1}}$ with a suitable variational lower bound that uses a 'critic' $U_t : \mathcal{H}^{t-1} \times \Theta \to \mathbb{R}$ to avoid explicit likelihood evaluations, where $\mathcal{H}^{t-1}$ and $\Theta$ are the spaces of histories and parameters respectively. An effective critic successfully encapsulates the true likelihood, tightening the bound. Although its form depends on the choice of bound, all critics are parametrized and trained in the same way, namely by a neural network $U_{\phi_t}$ which is optimized to tighten the bound. Unfortunately, replacing each $I_{h_{t-1}}$ involves learning $T$ such critic networks and requires samples from all posteriors $p(\theta|h_{t-1})$, which will typically be impractically costly.

To avoid this issue, we show that we can obtain a unified information objective similar to (4), *even without conditionally independent experiments*. The following proposition therefore marks the first key milestone in eliminating the restrictive assumptions of [17], by establishing a unified objective without intermediate posteriors that is valid even when the model itself changes between time steps.

**Proposition 1** (Generalized total expected information gain)**.** *Consider the data generating distribution $p(h_T|\theta,\pi) = \prod_{t=1:T} p(y_t|\theta,\xi_t,h_{t-1})$, where $\xi_t = \pi(h_{t-1})$ are the designs generated by the policy and, unlike in (4), $y_t$ is allowed to depend on the history $h_{t-1}$. Then we can write (3) as*

$$\mathcal{I}_T(\pi) = \mathbb{E}_{p(\theta)p(h_T|\theta,\pi)}\left[\log p(h_T|\theta,\pi)\right] - \mathbb{E}_{p(h_T|\pi)}\left[\log p(h_T|\pi)\right]. \tag{6}$$

Proofs are presented in Appendix A. The advantage of (6) is that we can draw samples from $p(\theta)p(h_T|\theta,\pi)$ simply by sampling our model and taking forward passes through the design network. However, neither of the *densities* $p(h_T|\theta,\pi)$ and $p(h_T|\pi)$ are tractable for implicit models.

To side-step this intractability, we observe that $\mathcal{I}_T(\pi)$ takes an analogous form to a MI between $\theta$ and $h_T$. For measure-theoretic reasons, namely because the $\xi_{1:T}$ are deterministic given $y_{1:T}$ (see Appendix A for a full discussion), it is not the true MI. However, the following two propositions show that we can treat $\mathcal{I}_T(\pi)$ *as if it were this MI*. Specifically, we show that the InfoNCE [57] and NWJ [37] bounds on the MI can be adapted to establish tractable lower bounds on our unified objective $\mathcal{I}_T(\pi)$. These two bounds both utilize a *single* auxiliary critic network $U_\psi$ that is trained simultaneously with the design network.

**Proposition 2** (NWJ bound for implicit policy-based BOED)**.** *For a design policy $\pi$ and a critic function $U : \mathcal{H}^T \times \Theta \to \mathbb{R}$, let*

$$\mathcal{L}_T^{NWJ}(\pi, U) := \mathbb{E}_{p(\theta)p(h_T|\theta,\pi)}\left[U(h_T,\theta)\right] - e^{-1}\mathbb{E}_{p(\theta)p(h_T|\pi)}\left[\exp(U(h_T,\theta))\right], \tag{7}$$

*then $\mathcal{I}_T(\pi) \geq \mathcal{L}_T^{NWJ}(\pi, U)$ holds for any $U$. Further, the inequality is tight for the optimal critic $U_{NWJ}^*(h_T,\theta) = \log p(h_T|\theta,\pi) - \log p(h_T|\pi) + 1$.*

**Algorithm 1:** Implicit Deep Adaptive Design with (iDAD)

---

**Input:** Differentiable simulator $f$, sampler for prior $p(\theta)$, number of experimental steps $T$
**Output:** Design network $\pi_\phi$, critic network $U_\psi$
**while** Computational training budget not exceeded **do**
    Sample $\theta \sim p(\theta)$ and set $h_0 = \varnothing$
    **for** $t = 1, ..., T$ **do**
        Compute $\xi_t = \pi_\phi(h_{t-1})$
        Sample $\varepsilon_t \sim p(\varepsilon)$ and compute $y_t = f(\varepsilon_t; \xi_t, \theta, h_{t-1})$
        Set $h_t = \{(\xi_1, y_1), ..., (\xi_t, y_t)\}$
    **end**
    Estimate $\nabla_{\phi,\psi} \mathcal{L}_T(\pi_\phi, U_\psi)$ as per (10) where $\mathcal{L}_T$ is $\mathcal{L}_T^{\text{NWJ}}$ (7) or $\mathcal{L}_T^{\text{NCE}}$ (8)
    Update the parameters $(\phi, \psi)$ using stochastic gradient ascent scheme
**end**
For deployment, use the deterministic trained design network $\pi_\phi$ to obtain a designs $\xi_t$ directly.

---

**Proposition 3** (InfoNCE bound for implicit policy-based BOED). *Let $\theta_{1:L} \sim p(\theta_{1:L}) = \prod_i p(\theta_i)$ be a set of contrastive samples where $L \geq 1$. For design policy $\pi$ and critic function $U : \mathcal{H}^T \times \Theta \to \mathbb{R}$, let*

$$\mathcal{L}_T^{NCE}(\pi, U; L) := \mathbb{E}_{p(\theta_0)p(h_T|\theta_0,\pi)} \mathbb{E}_{p(\theta_{1:L})} \left[ \log \frac{\exp(U(h_T, \theta_0))}{\frac{1}{L+1} \sum_{i=0}^L \exp(U(h_T, \theta_i))} \right], \tag{8}$$

*then $\mathcal{I}_T(\pi) \geq \mathcal{L}_T^{NCE}(\pi, U; L)$ for any $U$ and $L \geq 1$. Further, the optimal critic, $U_{NCE}^*(h_T, \theta) = \log p(h_T|\theta, \pi) + c(h_T)$ where $c(h_T)$ is any arbitrary function depending only on the history, recovers the sPCE bound in (5); the inequality is tight in the limit as $L \to \infty$ for this optimal critic.*

We propose these two alternative bounds due to their complementary properties: the NWJ bound can have large variance, but tends to be less biased. That is, the NWJ bound tends to be tighter for good critics, but is itself more difficult to reliably estimate and thus optimize. While the NWJ critic must learn to self-normalize, the InfoNCE bound avoids this issue but typically will not be tight for finite $L$ even with an optimal critic (note $\mathcal{L}_T^{\text{NCE}} \leq \log(L+1)$ [42]). Consequently, only the NWJ objective recovers the true optimal policy if our critic has infinite capacity and our optimization scheme is perfect, i.e. $\arg\max_\pi \max_U \mathcal{L}_T^{\text{NWJ}}(\pi, U) = \pi^* \neq \arg\max_\pi \max_U \mathcal{L}_T^{\text{NCE}}(\pi, U; L)$ in general, but it can be more difficult to work with in practice. We present a third bound that provides a potential solution to this, and further discuss the relative merits of the two bounds, in Appendix A.

We note that for both bounds the optimal critic does not depend on the learned policy. The final trained critic can be used to approximate the density ratio $p(h_T|\theta, \pi)/p(h_T|\pi) = p(\theta|h_T)/p(\theta)$, either directly in the case of the NWJ critic, or via self-normalization for the InfoNCE bound. We can use this to help approximate the posterior over $\theta$ given the collected real data from the experiment. This means we can perform likelihood-free inference after training the critic, which extends previous results [28, 29] from the static to the adaptive policy-based setting.

### 3.2 Parameterization and gradient estimation

In practice, we represent the policy $\pi$ and the critic $U$ as neural networks, $\pi_\phi$ and $U_\psi$ respectively, such that the lower bounds become a function $\mathcal{L}(\pi_\phi, U_\psi)$ of their parameters. By simultaneously optimizing $\mathcal{L}(\pi_\phi, U_\psi)$ with respect to both $\phi$ and $\psi$, we both learn a tight bound that accurately represents the true MI and a design policy network that produces high-quality designs under this metric.

We optimize these bounds using stochastic gradient methods [26, 49]. For this, we must account for the fact that the parameter $\phi$ affects the probability distributions with respect to which expectations are taken. We deal with this problem by utilizing the reparametrization trick [35, 48], for which we assume that design space $\Xi$ and observation space $\mathcal{Y}$ are continuous. To this end, we first formalize the notion of a differentiable implicit model in the adaptive design setting as

$$y_t = f(\varepsilon_t; \xi_t(h_{t-1}), \theta, h_{t-1}), \quad \text{where} \quad \theta \sim p(\theta), \quad \varepsilon_t \sim p(\varepsilon) \ \forall t \in \{1, \ldots, T\} \tag{9}$$

and we assume that we can compute the derivatives $\partial f/\partial \xi$ and $\partial f/\partial h$. Interestingly, it is possible to use an implicit prior without access to the density $p(\theta)$, and we do not need access to $\partial f/\partial \theta$.

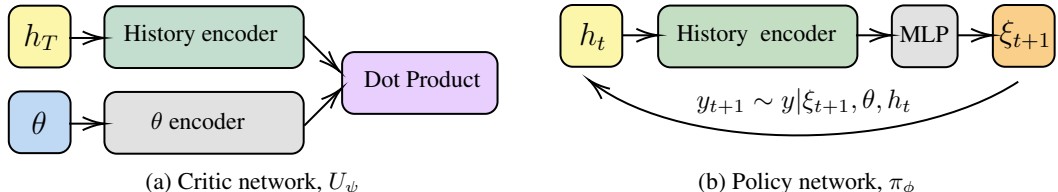

(a) Critic network, $U_\psi$

(b) Policy network, $\pi_\phi$

Figure 2: Overview of network architectures used in iDAD.

Under these conditions, we can express the bounds in terms of expectations that do not depend on $\phi$ or $\psi$, and hence move the gradient operator inside. For $\mathcal{L}_T^{\text{NCE}}(\pi_\phi, U_\psi; L)$, for example, we have

$$\nabla_{\phi,\psi}\mathcal{L}_T^{\text{NCE}} = \mathbb{E}_{p(\theta_{0:L})p(\varepsilon_{1:T})}\left[\nabla_{\phi,\psi}\log\frac{\exp(U_\psi(h_T(\varepsilon_{1:T},\pi_\phi),\theta_0))}{\frac{1}{L+1}\sum_{i=0}^{L}\exp(U_\psi(h_T(\varepsilon_{1:T},\pi_\phi),\theta_i))}\right]. \quad (10)$$

While each element of the history $h_T$ depends on $\phi$ in a possibly nested manner, we do not need to explicitly keep track of these dependencies thanks to automatic differentiation [5, 41].

Like DAD, our new method—which we call *implicit Deep Adaptive Design* (iDAD)—is trained with simulated histories $h_T = \{(\xi_i, y_i)\}_{i=1:T}$ prior to the actual experiment, allowing design decision to be made using a single forward pass during deployment. Unlike DAD, however, it does not require knowledge of the likelihood function, nor the assumption of conditionally independent designs, which significantly broadens its applicability. A summary of the iDAD approach is given in Algorithm 1.

### 3.3 Network architectures

The iDAD approach involves the simultaneous training of the *policy* $\pi_\phi$ and *critic* $U_\psi$ networks. It is essential to choose the neural architectures of these two components carefully to learn effective policies: poor choices of critic architecture will lead to loose, unrepresentative, bounds, while poor choices of policy architecture will directly lead to ineffective policies. Good choices of architecture need to balance flexibility with ease of training, and will typically require the incorporation of problem-specific inductive biases. A high-level summary of our architectures is shown in Figure 2.

The critic network, $U_\psi$, takes a *complete* history $h_T$ and the parameter $\theta$ as input, and outputs a scalar. Our suggested architecture first encodes the two inputs separately to representations of the same dimension, using a *history encoder*, $E_{\psi_h}$, and a *parameter encoder*, $E_{\psi_\theta}$, respectively. The output of the critic is then simply taken as their dot product $U_\psi(h_T, \theta) := E_{\psi_h}(h_T)^\top E_{\psi_\theta}(\theta)$; after training, the two encodings correspond to approximate sufficient statistics [9]. This setup corresponds to a separable critic architecture, as is commonly used in the representation learning literature [3, 8, 57]. While we use a simple MLP for $E_{\psi_\theta}$, the setup for $E_{\psi_h}$ varies with the context as we discuss below.

The policy network, $\pi_\phi$, takes the available history, $h_t$, as input, and outputs a design. Our suggested architecture makes use of a history encoder, $E_{\phi_h}$, of the same form as $E_{\psi_h}$, except that it must now take in varying length inputs; its output remains a fixed dimensional embedding. We then pass this embedding through an MLP to produce the design $\xi_{t+1}$. At the next iteration of the experiment, the same policy network is then called again with the updated history $h_{t+1} = h_t \cup \{(\xi_{t+1}, y_{t+1})\}$.

We use the same architecture for both history encoders, $E_{\psi_h}$ and $E_{\phi_h}$, but do not share network parameters between them. This architecture first individually embeds each design–outcome pair $(\xi_t, y_t)$ to a corresponding representation, $r_t$, using a simple MLP that is shared across all time steps. The produced history encoding is then an aggregation of these representations, with how this is done depending on whether the experiments are *conditionally independent*, i.e. $y_t \perp\!\!\!\perp h_{t-1}|\theta, \xi_t$, or not.

Foster et al. [17] proved that if experiments are conditionally independent, then the optimal policy is invariant to the order of the history. We prove that the same is true for the critic in Proposition 5 in Appendix C. In our setup, we can exploit this result by using a *permutation invariant* aggregation strategy for $\{r_1, \ldots, r_t\}$ when conditional independence holds. The simplest approach to do this would be to use sum-pooling [62], as was done in DAD. However, to improve on this, we instead propose using a more advanced permutation invariant architecture based on self-attention [13, 25, 39, 47, 59], namely that of Parmar et al. [40]; we find this provides notable empirical gains. When conditional independence does not hold, this approach is no longer appropriate and we instead use an LSTM [22] for the aggregation. See Appendix C for further details.

## 4 Related work

Adaptive policy-based BOED has only recently been introduced [17] and has not yet been extended to implicit models—the gap that this work addresses. Previous approaches to adaptive experiments usually follow the two-step greedy procedure described in Section 2. Methods for MI/EIG estimation without likelihoods include the use of variational bounds [15, 16, 28] and ratio estimation [27, 30]; approximate Bayesian computation together with kernel density estimation [43]; and approximating the intractable likelihood first, for example via polynomial chaos expansion [24], followed by applying likelihood-based estimators, such as nested Monte Carlo [46]. The maximization step in more traditional methods tends to rely on gradient-free optimization, including grid-search, evolutionary algorithms [44], Bayesian optimization [15, 30], or Gaussian process surrogates [38]. More recently, gradient-based approaches have been introduced [15, 28], some of which allow the estimation and optimization simultaneously in a single stochastic-gradient scheme [16, 23, 29]. From a posterior estimation perspective, likelihood-free inference can be performed via approximate Bayesian computation [33, 54], ratio estimation [56], conventional MCMC for methods that make tractable approximation to the likelihood [23, 24], or as a byproduct of MI estimation [16, 27, 29, 30].

## 5 Experiments

We evaluate the performance of **iDAD** on a number of real-world experimental design problems and a range of baselines. A summary of all the methods that we consider is given in Table 1. Since we aim to perform adaptive experiments in *real-time*, we focus mostly on baselines that do not require significant computational time during the experiment. These include heuristic approaches that require no training, namely **equal** interval designs (when possible) and **random**

Table 1: Key properties of considered methods.

|  | Adaptive | Real-time | Implicit |
|---|---|---|---|
| Random | ✗ | N/A | ✓ |
| Equal interval | ✗ | N/A | ✓ |
| MINEBED | ✗ | N/A | ✓ |
| SG-BOED | ✗ | N/A | ✓ |
| Variational | ✓ | ✗ | ✓ |
| DAD | ✓ | ✓ | ✗ |
| **iDAD** | ✓ | ✓ | ✓ |

designs, as well as static BOED approaches, where we, non-adaptively, choose all the designs prior to the experiment by optimising the mutual information objective of Equation (1) with $\xi = \{\xi_1, ..., \xi_T\}$ and $y = \{y_1, \ldots, y_T\}$. The static BOED approaches we consider are the **MINEBED** method of [28] and the likelihood-free ACE approach of [16], where we use the prior as a proposal distribution, referring to this baseline as **SG-BOED**. We also implement the expensive traditional non-amortized myopic strategy described in Section 2, for which we use the **variational** approach of [16], with the Barber-Agakov bound [4, 15], at each experiment step (see Appendix D.3 for details). Finally, where possible, we compare our method with DAD [17], in order to assess the performance gap that would arise if we had an analytic likelihood. This comparison is done primarily for evaluation purposes—because it has access to the likelihood density, DAD serves as an upper bound on the performance iDAD can achieve; one should use explicit likelihood methods whenever possible.

The main performance metric that we focus on is the total EIG, $\mathcal{I}_T(\pi)$, as given in (6). In cases where the likelihood is available, we estimate the $\mathcal{I}_T(\pi)$ using the sPCE lower bound in (5) and its sister upper bound, the sequential Nested Monte Carlo bound [sNMC; 17]. To ensure that the bounds are tight, we evaluate them with a large number of contrastive samples, i.e. $L \geq 10^5$. Where the likelihood is truly intractable, we assess the iDAD strategy in a more qualitative manner by looking at the optimal designs and approximate posteriors. For the adaptive experiments, we further consider the deployment time (i.e. the time required to propose a design), which is a critical metric for our aims. All deployment times exclude the time needed to determine the first experiment as it can be computed up-front, during the training phase. Timings for training the policy itself are given in Appendix D.

### 5.1 Location Finding

We first demonstrate our approach on the location finding experiment from [17]. Inspired by the acoustic energy attenuation model [53], this experiment involves finding the locations of multiple hidden sources, each emitting a signal with intensity that decreases according to the inverse-square law. The *total intensity*—a superposition of these signals—can be measured noisily at any location. The design problem is choosing where to measure the total signal in order to uncover the sources.

Table 2: Lower bounds on the total information, $\mathcal{I}_{10}(\pi)$, for the location finding experiment in Section 5.1. The bounds were estimated using $L = 5 \times 10^5$ contrastive samples. Errors indicate $\pm 1$ standard errors estimated over 4096 histories (128 for variational). Corresponding upper bounds are given in Table 6 in Appendix D.

| Method \ $\theta$ dim. | 4D | 6D | 10D | 20D |
|---|---|---|---|---|
| Random | $4.791 \pm 0.040$ | $3.468 \pm 0.014$ | $1.889 \pm 0.011$ | $0.552 \pm 0.006$ |
| MINEBED | $5.518 \pm 0.028$ | $4.221 \pm 0.028$ | $2.458 \pm 0.029$ | $0.801 \pm 0.019$ |
| SG-BOED | $5.547 \pm 0.028$ | $4.215 \pm 0.030$ | $2.454 \pm 0.029$ | $0.803 \pm 0.019$ |
| Variational | $4.639 \pm 0.144$ | $3.625 \pm 0.165$ | $2.181 \pm 0.151$ | $0.669 \pm 0.097$ |
| **iDAD** (NWJ) | $\mathbf{7.694 \pm 0.045}$ | $5.765 \pm 0.036$ | $\mathbf{3.252 \pm 0.039}$ | $\mathbf{0.877 \pm 0.022}$ |
| **iDAD** (InfoNCE) | $\mathbf{7.750 \pm 0.039}$ | $\mathbf{5.986 \pm 0.037}$ | $\mathbf{3.251 \pm 0.039}$ | $\mathbf{0.871 \pm 0.020}$ |
| DAD | $7.967 \pm 0.034$ | $6.300 \pm 0.030$ | $3.337 \pm 0.039$ | $0.937 \pm 0.022$ |

Table 3: Lower and upper bounds on MI $\mathcal{I}_{10}(\pi)$ for different network architectures on location finding experiment using the InfoNCE bound. All estimates obtained as in Table 2.

| Design | Critic | Lower bound | Upper bound |
|---|---|---|---|
| **Attention** | **Attention** | $\mathbf{7.750 \pm 0.039}$ | $\mathbf{7.863 \pm 0.043}$ |
| Attention | Pooling | $7.567 \pm 0.037$ | $7.632 \pm 0.039$ |
| Pooling | Attention | $7.398 \pm 0.040$ | $7.470 \pm 0.042$ |
| Pooling | Pooling | $7.135 \pm 0.034$ | $7.192 \pm 0.041$ |

In Table 2 we can see that iDAD substantially outperforms all baselines including, perhaps surprisingly, the traditional (non-amortized) adaptive variational approach, despite its large computational cost shown Table 4. The poor performance of the variational approach is likely driven by the inability of the mean-field variational family to capture the highly non-Gaussian true posterior, highlighting the detrimental effect that wrong posteriors can have on determining optimal designs when using the traditional sequential BOED approach.

Table 4: Deployment time of adaptive methods in 2D, measured on a CPU. Errors were calculated on the basis of 10 runs.

| Method | Deployment time (sec.) |
|---|---|
| Variational | $2256.0 \quad \pm 1\%$ |
| **iDAD** (NWJ) | $0.0167 \pm 2\%$ |
| **iDAD** (InfoNCE) | $0.0168 \pm 2\%$ |
| DAD | $0.0070 \pm 6\%$ |

Table 2 further shows that the performance gap to the likelihood-based DAD method is small, even as the dimension of the design and parameter space grows. Though the information gained by all methods decreases with the dimensionality, this is to be expected: in higher dimensions it is inherently more difficult to infer the relative direction of the sources from observing their intensity. Overall, this experiment demonstrates that iDAD is able to learn near-optimal amortized design policies without likelihoods, while being run in milliseconds at deployment.

**Ablation: attention to history.** We next assess the benefit of utilizing our proposed more sophisticated permutation invariant architectures, compared to the simple pooling of [62] used in [17]. Our approach incorporates attention layers into both networks that we train. This leads us to four possible combinations of network architectures. Table 3 compares the efficacy of the resulting design policies and strongly suggests that incorporating attention mechanisms in either and/or both networks improves performance, with inclusion in the design network particularly important.

We preform further ablation studies to investigate and demonstrate important properties of our method, such as its scalability with the number of experiments $T$, stability between different training runs and performance to errors in the design network (introduced by not training the network to convergence). Results and discussion are provided in Appendix D.4.4.

## 5.2 Pharmacokinetic model

Our next experiment is taken from the pharmacokinetics literature and has been studied in other recent works on BOED for implicit models [28, 63]. Specifically, we consider the compartmental model

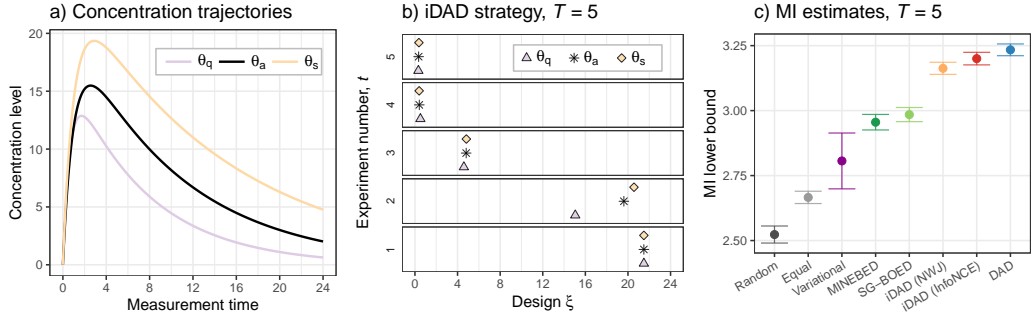

Figure 3: Plots for pharmacokinetics experiment. a) Visualisation of model showing concentration level as a function of measurement time for 3 values of $\theta$, resulting in a quick ($\theta_q$), average ($\theta_a$), or slow ($\theta_s$) trajectory. b) Designs selected by an iDAD policy trained with InfoNCE. c) MI lower bounds achieved by iDAD and baselines. All estimates obtained as in Table 2.

of [50], for which the distribution of an administered drug through the body is governed by three parameters: absorption rate $k_\alpha$, elimination rate $k_e$, and volume $V$, which form the parameters of interest, i.e. $\theta = (k_\alpha, k_e, V)$. Given $T = 5$ patients, the design problem is to adaptively choose blood sampling times, $0 \le \xi_t \le 24$ hours, for each, measured from the the point the drug was administered (with patient 2 not being administered until after sampling patient 1 etc). Plausible concentration trajectories are shown in Figure 3a). Full details and further results are given in Appendix D.5.

We first qualitatively consider the design policy of iDAD (trained with the InfoNCE objective) in Figure 3b). As we have not yet observed any data, the optimal design for the first patient (bottom row) is the same for all $\theta$. For the second patient, only guided by $\xi_1$ and the outcome $y_1$, iDAD is already able to distinguish between quickly and slowly decaying concentration trajectories: it proposes a significantly earlier measurement time for the quickly decaying trajectory (purple triangle, $\theta_q$) and later time for the slowly decaying one (yellow diamond, $\theta_s$). For the third patient, iDAD always targets the peak of the drug concentration distribution which is quite similar for all $\theta$. Measurements for the last two patients are made soon after the drug has been administered ($\sim 15 - 30$ min), when concentration levels increase rapidly, to capture information about how quickly the drug is absorbed.

To provide more quantitative assessment and compare to our baselines, we again consider the final EIG values as shown in Figure 3c). This reveals that the iDAD strategies perform best among the methods that are applicable to implicit models, confirming that the learnt policies propose superior designs. The performance gap to DAD, which relies on explicit likelihoods, is not statistically significant (at the 5% level) for iDAD trained with InfoNCE, while significant, but still small, for NWJ.

Finally, we consider the convergence of the iDAD networks under the different training objectives and compare to DAD for reference. As shown in Figure 4, although all three converge to approximately the same value, they do so at rather different speeds: while DAD requires about 5000 gradient updates, implicit methods need longer training and tend to exhibit higher variance, particularly NWJ.

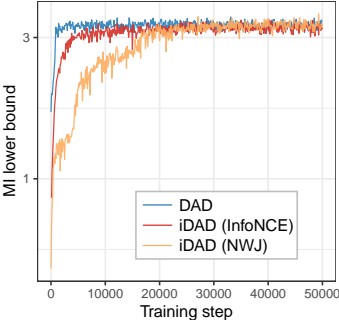

Figure 4: Convergence of MI lower bounds.

## 5.3 SIR Model

In this experiment, we demonstrate our approach on an implicit model from epidemiology. Namely, we consider a formulation of the stochastic SIR model [10] that is based on stochastic differential equations (SDEs), as done by [29]. Here, individuals in a fixed population belong to one of three categories: susceptible, infected or recovered. Susceptible people can become infected and then recover, with the dynamics of these two events being governed by two model parameters—the infection rate $\beta$ and the recovery rate $\gamma$. Our aim is to determine the optimal times $\tau$ at which to measure the number of infected people, $I(\tau)$, in order to estimate the two parameters. This implicit model is challenging because data simulation is expensive, since we need to solve many SDEs, and experimental designs have a time-dependency. See Appendix D.6 for full details.

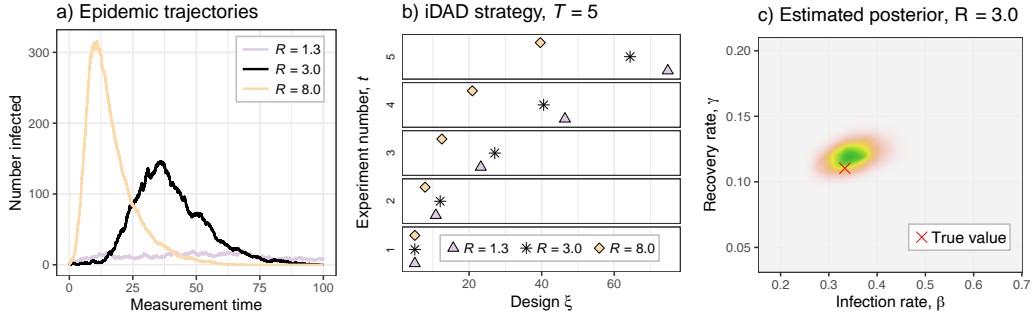

Figure 5: a) Epidemic trajectories for 3 realization of $(\beta, \gamma)$ with different reproduction numbers $R = \beta/\gamma$. b) Designs selected by an iDAD policy trained with NWJ. c) Example posterior estimates from the critic network given data generated with the ground-truth parameters shown by the red cross.

We train a iDAD networks to perform $T = 5$ experiments and compare against random, equal interval, and static design baselines; DAD cannot be run because the problem corresponds to a true implicit model. Table 5 shows lower bound estimates on the MI and demonstrates that iDAD outperforms all compared methods. Note that a degree of caution is required when analysing the results, as they are influenced by unavoidable biases in the estimation process. Namely, a critic is still required to estimate the MI lower bound, and there may be variations in the effectiveness of these critics, with less effective ones corresponding to looser bounds and therefore underestimating the true MI. Nonetheless, for other models where such checks are possible, we have found the bounds to be relatively tight, while, even if this turns out not to be the case here, the fact that the critics for the static approaches are easier to train should mean our relative evaluations for iDAD (and Random) are still conservative compared to the other baselines.

Table 5: MI lower bounds ($\pm 1$ s.e.).

| Method | Lower bound |
|---|---|
| Random | $1.915 \pm 0.032$ |
| Equal interval | $2.669 \pm 0.023$ |
| MINEBED | $3.400 \pm 0.001$ |
| SG-BOED | $3.752 \pm 0.020$ |
| **iDAD** (NWJ) | $\mathbf{3.869 \pm 0.001}$ |
| **iDAD** (InfoNCE) | $\mathbf{3.915 \pm 0.020}$ |

Figure 5 further demonstrates important qualitative results for this model. Figure 5a) shows different epidemic trajectories, i.e. the number of infected $I(\tau)$ people as a function of measurement time $\tau$, whilst 5b) plots their corresponding designs obtained from the learned iDAD policy. Importantly, diseases with a significantly different profile, e.g. a slow ($R = 1.3$) or a fast ($R = 8.0$) spread result in different sets of optimal designs, highlighting the adaptivity of iDAD. Finally, Figure 5c) shows an example posterior distribution estimate from the learnt iDAD critic network, which we see is consistent with the ground truth parameters.

## 6 Discussion

**Limitations.** The benefit that iDAD can be used in live experiments comes at the cost of substantial training that can be computationally expensive. However, this is mitigated by its amortization of the adaptive design process, such that only one network needs training, even if we have multiple experiment instances. The cost–performance trade-off can also be directly controlled by judicious choices of architecture and the amount of training performed. Another natural limitation is that the use of gradients naturally restricts the approach to continuous design settings, something which future work might look to address.

**Conclusions.** In this paper we introduced iDAD—the first policy-based adaptive BOED method that can be applied to implicit models. By training a design network without likelihoods upfront, iDAD is thus the first method that allows real-time adaptive experiments for simulator-based models. In our experiments, iDAD performed significantly better than all likelihood-free baselines. Further, by using models where the likelihood is available as a test bed, we found that it was able to almost match the analogous likelihood-based adaptive approach, which acts as an upper bound on what might be achieved without access to the likelihood itself. In conclusion, we believe iDAD marks a step change in Bayesian experimental design for *implicit* models, allowing designs to be proposed quickly, adaptively, and non-myopically during the live experiment.

## Acknowledgments and Disclosure of Funding

DRI is supported by EPSRC through the Modern Statistics and Statistical Machine Learning (StatML) CDT programme, grant no. EP/S023151/1. AF gratefully acknowledges funding from EPSRC grant no. EP/N509711/1. SK was supported in part by the EPSRC Centre for Doctoral Training in Data Science, funded by the UK Engineering and Physical Sciences Research Council (grant EP/L016427/1) and the University of Edinburgh.

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
