# OpenReview forum: "Implicit Deep Adaptive Design: Policy-Based Experimental Design without Likelihoods"
_NeurIPS.cc/2021/Conference — NeurIPS 2021 Poster_

### Official Review · Reviewer_8e2K · 2021-07-16

**Rating:** 7
**Confidence:** 3

**Summary:**

In this work, the authors extend the DAD method by using implicit models to overcome some weaknesses of DAD. The authors further derive several bounds to design re-parametrization gradients. The authors show that the proposed method outperforms existing methods in several Bayesian experimental design tasks.

**Limitations And Societal Impact:**

yes

**Main Review:**

Originality: It is new to use implicit models in the setting of Bayesian experimental design. This work is a novel combination of implicit models and DAD.

Quality: It seems that the proposed method is technically sound and promising. I do not check the correctness of the proofs in the appendix. The experimental results also support the claim.

Clarity: This submission is easy to read and well-organized.

Significance: I think this work extends the scope of DAD and it could be useful for Bayesian experimental design tasks.

**Time Spent Reviewing:**

3

---

> ### Author Response · Authors · 2021-08-10
> **Individual response to Reviewer 8e2K**
>
> Thank you for your review. We are very pleased that you consider our proposed method to be technically sound and promising. We fully agree that iDAD vastly expands the model space for policy-based Bayesian experimental design: it constitutes the first method in the literature that can practically perform real-time adaptive experiments with implicit models. Previous approaches are either not fast enough to run in real-time (lines 93-95) or require explicit likelihood models (lines 111-114).

---

### Official Review · Reviewer_B4Q3 · 2021-07-16

**Rating:** 7
**Confidence:** 3

**Summary:**

The paper tackles the problem of adaptive Bayesian experimental design. Specifically, the authors try to relax the conditionally independent experiments and the explicit likelihood of the policy-based adaptive design model proposed by Foster. Code and supplemental materials are provided.

There are 3 contributions:
- By leveraging the advance in variational mutual information, the authors can establish a suitable likelihood-free objective in the implicit setting.
- The authors theoretically show that they can obtain a unified information objective without conditionally independent experiments.
- Based on those insights, the authors propose the iDAD algorithm, which is much based on the previous DAD work.


**Limitations And Societal Impact:**

The authors did not provide limitations and societal impact directly derived from this work.

**Main Review:**

Main Strengths:
- The paper makes a very good connection between the difficulties and limitations of the previous  DAD models and how they derive the idea of new modifications. This shows a very clear motivation of the authors in this paper.
- The idea of using a critic to derive a lower bound for the objective function and optimize this lower bound at the same time with the objective function is interesting, somewhat similar to the development of actor-critic from Reinforce. This mechanism helps to extend iDAD to various applications.
- The authors provide a solid theoretical foundation for their proposed method.  The author successfully eliminates conditional independence which is a restriction of the previous DAD model.
- All the proofs are clear and easy to understand although the idea of the proofs is not new. Overall the paper is well presented and easy to follow; the structure of the paper is similar to DAD (which is not a bad thing).

Weakness:
- The method requires a lot of pre-training which can be computationally expensive.
- The performance of iDAD is still worse than that of DAD in many experiments shown. Probably the authors need to reserve more space to do more ablation and analyze why it is the case. For example, when claiming there is a “cost-performance trade-off”, we also want to see when the performances peak, do they beat that of DAD and why?


**Time Spent Reviewing:**

10

---

> ### Author Response · Authors · 2021-08-10
> **Individual response to Reviewer B4Q3**
>
> Thank you for your review. We are very glad to hear that you found our work well-presented, well-motivated and easy to follow, including the proofs—thanks for reading through them! We hope the detailed responses below address the questions you have.
>
>
> > The method requires a lot of pre-training which can be computationally expensive
>
> Though this is true, as mentioned in the paper (line 358-359), we emphasize that one usually has far more time available *before* than *during* the experiment.  Thus iDAD invests in training upfront, before the start of the experiment, to get very quick and adaptive design decisions during deployment when time is at a premium.
>
> Furthermore, once trained iDAD can be applied across multiple experiment instances without additional training cost, unlike non-amortized adaptive methods. For instance, on the location finding experiment with just 20 repeats of the experiment, iDAD consumes less time in total (training plus deployment) than the variational baseline, because the non-amortized baseline starts afresh for each repeat. Thus, the training cost of iDAD is very modest compared to, for example, inference amortization, which often requires many orders of magnitude more compute than non-amortized inference. Finally, we note that non-adaptive methods such as MINEBED and SG-BOED also require training prior to the experiment.
>
> > The performance of iDAD is still worse than that of DAD in many experiments shown.
>
> We feel there might be a misunderstanding here: DAD simply does not work for the core models that iDAD is designed to deal with, such as time-series models and SDE models. The SIR model (third experiment) is precisely such a model. Therefore, we would like to stress that iDAD and DAD (or other non-implicit methods) are not competing approaches: DAD assumes access to the likelihood function and conditionally independent designs, whilst iDAD is introduced to relax these requirements, not to improve performance when the requirements are already met. We tested iDAD on explicit likelihood models (first two experiments) so that we could calibrate its performance, in a manner analogous to doing experiments with a known ground truth that can be compared to. In more detail, in these scenarios, DAD represents an *upper bound* on the performance iDAD can achieve (see lines 258-262). Therefore, it is a big win for iDAD to come within 3% of the DAD result in the Location Finding model (section 5.1) and within 1% for the Pharmacokinetic model (section 5.2).
>
>
> > For example, when claiming there is a “cost-performance trade-off”, we also want to see when the performances peak, do they beat that of DAD and why?
>
> The cost-performance trade-off we refer to in the discussion is between the computational cost of training the iDAD networks and the performance of the policy at test time. Longer training results in better performance (there is no peak). However, the returns on training are diminishing. We investigate this in the ablation study in response to Reviewer k7ct, where Table B shows that with 2% of the total budget we achieve around 80% of the performance of the fully trained network; 8% of the total budget gets us to around 90%, and that number continues to slowly increase as we continue training.
>
> Regarding the question if we can beat DAD and why: iDAD is designed to cover the common scenarios where DAD cannot be applied (as discussed above), we are not recommending its use when DAD is applicable (as DAD should always be better when you can actually use it, see lines 258-262).

---

### Official Review · Reviewer_k7ct · 2021-07-17

**Rating:** 7
**Confidence:** 4

**Summary:**

This paper proposes iDAD, an amortized learning framework for implicit (likelihood-free) experimental design. The iDAD framework extends the previously proposed deep adaptive design (DAD) method (Foster et al., 2013), which is an amortized learning framework for experiment design in likelihood-based (i.e. non-implicit) models, which learns a policy network offline that can then be deployed online for fast experimental design. Additionally, the proposed iDAD framework aims to alleviate the constraint in DAD that requires conditionally independent experiments.

**Limitations And Societal Impact:**

This paper includes a section on limitations in the Discussion section, which is a nice aspect of the paper.

**Main Review:**

This paper is written quite clearly and does a good job in providing context for the proposed iDAD method. As such, the contributions of this paper are easy to follow. However, there are a few places where I think this paper could have provided additional details or empirical study, which I summarize below.

I would appreciate more discussion and experimental evaluation on the cost/difficulty of training the policy network to an accurate level under a greater variety of settings. For example, are there conditions, in terms of dimensionality of the input space, dimensionality of the parameter space, query budget or other factors, in which the cost of training an accurate network becomes prohibitive? It would be ideal to include an ablation study that investigates the settings where training becomes difficult, and show the rate at which performance suffers in these conditions, relative to non-amortized methods.

Related to this, I think this paper would benefit from including a discussion (or even some experimental evaluation) on situations where there are (slight) errors with the design policy network—i.e. showing how much will minor errors affect the performance of this amortized method in practice, and detailing any strategies to mitigate any issues caused by these errors.

The iDAD method seems to still have the restriction that one is able to compute the derivative of y with respect to the design point (e.g. via automatic differentiation). While there were a few examples shown of likelihood-free BOED situations where this is possible, it would be great to try and better characterize the scope or types of models that this applies to, and any examples of cases where the framework is limited due to this requirement.

Finally, as the proposed work is motivated by the existing DAD method, I feel that this paper would be improved if there were more specific examples in the introduction of implicit BOED settings where the proposed methods provide advantages over the DAD (and other non-implicit BOED) methods.

—————————————————

**Update:** I thank the authors for their detailed response to my review, and for providing the additional empirical results. I believe these have sufficiently answered all of my questions and concerns that I described above.

**Time Spent Reviewing:**

2

---

> ### Author Response · Authors · 2021-08-10
> **Individual response to Reviewer k7ct**
>
> Thank you for your review. We are very pleased that you found our work clear and that the contributions we make are easy to follow. We hope the detailed responses below address the questions that you have.
>
>
> > I would appreciate more discussion and experimental evaluation on the cost/difficulty of training the policy network to an accurate level under a greater variety of settings.
>
> Thanks for this question—this is a great idea; we ran an ablation study with the location finding model (section 5.1 in the paper) to investigate the scalability of iDAD. Specifically, we train iDAD networks to perform T=10 experiments to locate 2 sources, varying the dimensionality of the input space. This scales up the dimension of both the design and the parameter space.
>
> Table A shows results for 6-, 10- and 20-dimensional parameter space, whilst the dimension of the design variable is 3, 5 and 10 per experiment, which gives a total of 30, 50 and 100 designs across the 10 steps of the experiment. In addition to iDAD, we include a random baseline and DAD. (We did not have time to run static and variational baselines, but we will include them when revising the paper).
>
> As a simple measure of training difficulty, we look at the performance gap between iDAD and DAD (which acts as an upper bound on performance). We show this with % in parentheses. Based on that metric, we see that iDAD scales well with dimensions, as it is able to come within 10% of DAD’s performance. We further note that iDAD comfortably outperforms the random design baseline across the board by a margin of above 50%.
>
> We note that we used the same network architecture to obtain the results in Table A. Therefore we expect to see performance improvement if the network sizes are scaled with the dimensionality of the problem.
>
> _**Table A**: Ablation study on the scalability of iDAD for the location finding experiment in higher dimensions. The table reports mutual information lower bounds and performance gap between iDAD and DAD in parentheses. The size of the networks was held constant across all dimensions._
>
>
> | Method \ $\theta$ dimension  | 4D (from paper) |       6D        |      10D       |     20D         |
> | --------------------------------------- | ---------------------- | -------------- | --------------- | ---------------- |
> | DAD                                        | 7.967                   | 6.300         | 3.338          | 0.937           |
> | iDAD                                       | 7.750 (3%)          | 5.986 (5%) | 3.251 (3%) |  0.871 (7%)  |
> | iDAD without attention           |    7.135                 | 5.375         | 2.677          | 0.695      |
> |Random design baseline         |  4.791                  |  3.468        | 1.889          | 0.552           |
>
>
> > For example, are there conditions, in terms of dimensionality of the input space, dimensionality of the parameter space, query budget or other factors, in which the cost of training an accurate network becomes prohibitive?
>
> If the networks parametrizing the policy and the critic are not expressive enough, scaling to high dimensions might prove more difficult. For example, using the simpler pooling architecture (“iDAD without attention” in Table A) delivers performance that lags behind both iDAD and DAD as dimensionality increases.
>
> It is worth highlighting that the dimensions of both the parameter and the design space of the models usually studied in the BOED literature are relatively low, typically $<5$ in the implicit model setting. Therefore, in the context of implicit BOED, 20-dimensional parameter spaces and 100-dimensional designs can be considered as high-dimensional.
>
>
>
> > Related to this, I think this paper would benefit from including a discussion (or even some experimental evaluation) on situations where there are (slight) errors with the design policy network—i.e. showing how much will minor errors affect the performance of this amortized method in practice [...]
>
> This is an excellent suggestion for another ablation study, which we again ran on the location finding model. Table B shows the performance of iDAD as a function of training time. The idea is that, before convergence, some errors or inaccuracies are present in the policy, so we can use a partially trained network to investigate the effect of slight errors in the design policy network. We find that small errors in the network only lead to small drops in performance.
>
> In detail, our results show that with just 8% of the total training budget, this slightly inaccurate network still performs relatively well, achieving total mutual information of 7.1, compared to the fully trained network that reached 7.8.  We also highlight that iDAD outperforms all baselines with as little as 1% of the total training budget (the best performing baseline achieves mutual information of 5.5, see Table 2 in the paper).
>
>
> _**Table B**: Ablation study on the performance of iDAD as a function of training time for the location finding experiment._
>
>
> | % of training budget | MI lower bound |
> |----------------------: |----------------: |
> |                 0.1% |          3.38 |
> |                 1.0% |          6.09 |
> |                 2.0% |          6.46 |
> |                 4.0% |          6.81 |
> |                 8.0% |            7.08 |
> |                16.0% |             7.33 |
> |                32.0% |            7.56 |
> |                64.0% |            7.78 |
> |               100.0% |               7.82 |
>
>
> > [...] detailing any strategies to mitigate any issues caused by these errors.
>
> A useful property of iDAD is that many potential issues can be diagnosed and resolved before actual deployment. For example, one could qualitatively evaluate the designs where possible (e.g. as we do in Figure 1b, lines 306-314), ensuring that these “make sense” for the model at hand. Quantitative assessment versus simple baselines that do not require training (such as random designs) can be beneficial in diagnosing training/optimization problems. If such issues do arise, we can apply a wide range of deep learning techniques known to be helpful in that regard (e.g. batch norm, use of bigger networks and improved architectures).
>
> We will include such a discussion when revising the paper, and we thank you again for this suggestion.
>
> > Better characterize the scope or types of models that this applies to, and any examples of cases where the framework is limited due to this requirement.
>
> Thanks for this point. Indeed, iDAD requires the underlying process to be differentiable and whilst we specify the kinds of models iDAD is suitable for (in the abstract, the intro lines 45-49, gradient section lines 185-188 and lines 362-363 of the discussion), more concrete examples are a good idea. Here are several more examples, which we will include in the paper too.
>
> - Beyond the examples mentioned in the introduction, namely outbreak models in epidemiology, models from ecology and chemistry (lines 48-49), further examples include: models of neural dynamics in neuroscience, many models specified via stochastic differential equations, mixed effects and random effects models, goal-oriented design problems (where the target is a function of a larger set of parameters), particle simulator models, and graphics renderers. Further, if a differentiable neural network emulator can be learned for a non-differentiable generative model (Lueckmann et al., 2019, Nonnenmacher and Greenberg, 2021), then iDAD can be applied using the emulator.
>
> - Examples where iDAD does not apply include: situations where the design variable is discrete or, more generally, non-differentiable generative processes which cannot be faithfully approximated with a differentiable process. We note though that there are currently no scalable BOED approaches in the literature for general non-differentiable implicit models.
>
>
> > I feel that this paper would be improved if there were more specific examples in the introduction of implicit BOED settings where the proposed methods provide advantages over the DAD (and other non-implicit BOED) methods
>
> The most significant advantage of iDAD over methods that require an explicit likelihood is its much broader applicability. For example, models specified via stochastic differential equations (SDEs) do not typically have an explicit likelihood function; SDEs are ubiquitous across scientific disciplines and are in the scope of models that iDAD applies to, but DAD does not. Again—thanks for this point; we appreciate that including more specific examples is a good idea and will make sure we do when revising the paper.
>
> ---
> References:
>
> Lueckmann JM, Bassetto G, Karaletsos T, Macke JH. Likelihood-free inference with emulator networks. In Symposium on Advances in Approximate Bayesian Inference, 2019 (pp. 32-53). PMLR.
>
> Nonnenmacher, Marcel and Greenberg, David S. Deep Emulators for Differentiation, Forecasting, and Parameterization in Earth Science Simulators. Journal of Advances in Modeling Earth Systems, 2021, volume 13 (7).

---

### Official Review · Reviewer_nc49 · 2021-07-19

**Rating:** 7
**Confidence:** 1

**Summary:**

This paper proposes a method for Bayesian Optimal Experiment Design (BOED) which uses a policy network to decide which experiment to perform next based on the history of past experiments and their outcomes. The goal is to enable experiments to be chosen quickly for real-time applications. The method trains a policy network in simulation, and does not require assumptions necessary for a prior method (DAD), which also trains a policy network in simulation but requires additional assumptions such as the availability of a likelihood model of the history given the parameters and policy. The present approach removes this requirement by using variational approximations together with a reparameterization which enables optimization with gradient descent. Experiments are done in a 2D location finding environment, a pharmacokinetics model and an epidemiology simulation. Results show that the proposed method almost matches the performance of an exact method which has access to exact likelihood on the first task, and does better than baselines on all the tasks while being fast to evaluate.

**Limitations And Societal Impact:**

Yes, they mention the substantial computational cost of pretraining.

**Main Review:**

This paper is very much outside my research area and I’m not familiar with the current state of the art in the area so my review is an educated guess. From what I can tell the paper addresses an important question, which is being able to perform experiments in real time (they give the example of tracking the source of a chemical spill which I found to be a good motivating example) for a general class of problems which only require the ability to simulate from the environment and compute gradients of the measurement wrt design. This is still not totally general since not all processes are differentiable, which should be mentioned somewhere The paper is very well written and did a good job of making the main arguments accessible.

**Time Spent Reviewing:**

2.5

---

> ### Author Response · Authors · 2021-08-10
> **Individual response to Reviewer nc49**
>
> Thank you for your review. We are glad to hear that our paper was accessible and well written. To aid in your confidence on your assessment, we agree that you identified two of the key strengths of work:
>
> 1. The proposed method (iDAD) broadly matches the performance of methods that additionally assume access to the likelihood function, i.e. Deep Adaptive Design (DAD) by Foster et al. 2021. This is far from trivial since the likelihood plays a crucial role in the expected information gain.
>
> 2. The proposed method does better than previous approaches that do not assume access to the likelihood function while being faster to evaluate, enabling real-time experimentation with numerous applications.
>
> > This is still not totally general since not all processes are differentiable, which should be mentioned somewhere
>
>
> Thanks for highlighting this point. We mention the assumption on differentiability (in the abstract, the intro lines 45-49, gradient section lines 185-188 and lines 362-363 of the discussion), but we will make it clearer.
>
> We emphasize that differentiable implicit models constitute a large proportion of implicit models (largely thanks to the rise in popularity of automatic differentiation). At the same time, we note that there are currently no scalable BOED approaches in the literature for general non-differentiable implicit models.

---

### Author Response · Authors · 2021-08-10
**Response to all reviewers and additional experiments**

We thank all the reviewers for their hard work, thoughtful feedback and comments. We were very pleased to see our submission described as clearly motivated (B4Q3), well-written and accessible (nc49), well presented (B4Q3) and well organised (8e2K), providing a clear context for the proposed method (k7ct).  Reviewers thought the novel contributions of the paper were clear and easy to follow (k7ct, B4Q3) and that the  proposed method  was technically sound (8e2K) with solid theoretical foundations (B4Q3).

To further enhance the paper and answer questions from the reviewers (particularly k7ct), we have run several new ablation studies to understand why iDAD performs as well as it does. We investigated questions such as the scalability of our method with dimension and sensitivity of performance to (small) errors in the design policy.

To briefly summarize the results: we find that iDAD scales well with the dimensionality of the parameter and design space, although training can become difficult if the design and critic networks are not expressive enough (Table A). Finally, we find that small errors in the design policy (introduced by not training the network to convergence) only lead to small decreases in performance (Table B). We believe that these ablation studies further strengthen the experimental results in our work and will include them when revising the paper.

---

### Decision · Program_Chairs · 2021-09-27

**Decision:**

Accept (Poster)

**Comment:**

The paper is on Bayesian experimental design, building on the recently proposed Deep Adaptive Design (DAD). The paper's contribution is an extension iDAD (I for implicit) that does not require explicit likelihoods, does not require that experiments are conditionally independent, and is fast enough for real-time applications. The paper is interesting and thorough.